

# Fecal microbiota transplantation research output from 2004 to 2017: a bibliometric analysis

Yan Li[1,2,*], Ziyuan Zou[3,*], Xiaohui Bian[2], Yushan Huang[2], Yanru Wang[4], Chen Yang[3], Jian Zhao[2] and Lang Xie[1]

[1] Department of General Surgery, Zhujiang Hospital of Southern Medical University, Guangzhou, Guangdong, China

[2] The Second School of Clinical Medicine, Southern Medical University, Guangzhou, Guangdong, China

[3] The First School of Clinical Medicine, Southern Medical University, Guangzhou, Guangdong, China

[4] Department of Radiation Oncology, Zhujiang Hospital, Southern Medical University, Guangzhou, Guangzhou, China

[*] These authors contributed equally to this work.

Corresponding author
Lang Xie, langxiezj@hotmail.com

## ABSTRACT

**Background**. Fecal microbiota transplantation (FMT) is an emerging therapy against *Clostridium difficile* infection (CDI) and inflammatory bowel disease (IBD). Although the therapy has gained prominence, there has been no bibliometric analysis of FMT.
**Methods**. Studies published from 2004 to 2017 were extracted from the Science Citation Index Expanded. Bibliometric analysis was used to evaluate the number or cooperation network of publications, countries, citations, references, journals, authors, institutions and keywords.
**Results**. A total of 796 items were included, showing an increasing trend annually. Publications mainly came from 10 countries, led by the US ($n = 363$). In the top 100 articles ranked by the number of citations (range 47–1,158), American Journal of Gastroenterology (2017 IF = 10.231) took the top spot. The co-citation network had 7 co-citation clusters headed by 'recurrent *Clostridium difficile* infection'. The top 7 keywords with the strongest citation bursts had three parts, 'microbiota', ' diarrhea ', and 'case series'. All keywords were divided into four domains, 'disease', 'nosogenesis', 'trial', and 'therapy'.
**Conclusions**. This study shows the research performance of FMT from 2004 to 2017 and helps investigators master the trend of FMT, which is also an ongoing hotspot of research.

## INTRODUCTION

Fecal microbiota transplantation (FMT) is a therapeutic method by infusing fecal suspensions from a healthy individual into the gastrointestinal tract (*Kelly et al., 2016*). *Zhang et al., (2012)* noted that GE Hong pioneered the use of feces to treat human diseases in the Eastern Jin Dynasty (300∼400 AD) FMT has received public attention over the past decade because of its highly effective treatment of *Clostridium difficile* infection (CDI) and

inflammatory bowel disease (IBD) (*Khoruts et al., 2016*; *Khoruts, Sadowsky & Hamilton, 2015*). In 2013, FMT was included in the CDI treatment guidelines which clearly stipulated that FMT should be considered for patients with a third recurrence of CDI (*Surawicz et al., 2013*). It was pointed out that recurrent CDI was difficult to treat, and the failure rate of antibiotic therapy was relatively high. Moreover, it has been reported that more than 300 cases of recurrent CDI were effectively treated using FMT (*Van Nood et al., 2013*). FMT not only has a remarkable cure rate, but is also a safe and acceptable treatment option (*Brandt et al., 2012*). However, little systematic analysis of FMT has been performed (*Hourigan et al., 2015*).

Systematic reviews and meta-analyses are being increasingly used to summarize medical literature and identify areas in which research is needed (*Crowther, Lim & Crowther, 2010*). Unfortunately, they fail to include all the relevant research; meanwhile, the definition of clinical endpoint for the use in combined statistics is unclear and they have publication bias. Quantitative studies of the literature have been performed for nearly one hundred years, during which bibliometric methods have been developed and matured (*Rousseau, 2014*). Traditional bibliometric methods had been used to evaluate the variations of particular areas by assessing the productivity of countries, institutions, authors and journals (*Keathley-Herring et al., 2016*). Nowadays, bibliometric analysis provides a statistical and quantitative analysis of publications and offer a convenient way to visibly measure researchers' efforts in the investigation of a specific field (*Ashok et al., 2016*; *Yao et al., 2018*).

In the current study, we analyzed those were quoted above and provided additional analysis including keywords of those studies and the clusters of citations (*Choi & Kim, 2018*; *Miao et al., 2017*; *Suk et al., 2011*). Multiple analytical tools were used to map the trends of FMT research from 2004 to 2017. This analysis will assist researches in understanding the literature regarding FMT, and determining the future directions for future study of FMT.

## MATERIALS & METHODS

### Search strategy

Data were acquired from the Science Citation Index Expanded (SCI-E) of the Web of Science Core Collection (WoSCC) of Clarivate Analytics (https://clarivate.com/) on April 22, 2018. The data were downloaded from the WoSCC so that there were no ethical issues. The searching included literature published from 2004 to 2017, and used the following keywords and terms: 'Fecal bacteria transplantation$' or 'Intestinal Microbiota Transfer$' or 'Fecal Transplantation$' or 'Fecal Transplant$' or 'Donor Feces Infusion$'. All electronic searches were performed on the same day, April 22, 2018, to avoid changes in citation rates. The year 2018 was excluded because database entries for the year would not be complete at the time of the search. When all data were collected, the results were arranged according to the 'Times cited'.

### Study selection

Two independent reviewers (Y Li and YS Huang) collected all the data by reading the titles and abstracts acquired from SCI-E of the WoSCC database. When necessary, the full text was downloaded from PubMed or other databases. Articles were included only if the main

topic was FMT, and the language was English. Exclusion criteria were: (1) the main topic of the article was not about FMT; (2) the abstract of the article couldn't be acquired from WoSCC; (3) the article was a duplicate. Any differences between the two reviewers were settled through discussion with a third reviewer.

## Assessment of the articles and journals

Two researchers (YS Huang and J Zhao) reviewed the selected articles, and the following data were identified and recorded for analysis: (1) titles, (2) authors, (3) citation number, (4) keywords, (5) publication year, (6) topics (7) funding and (8) countries of origin. Furthermore, the journal names and impact factors (IFs) were also recorded using the 2017 edition of the Journal Citation Reports (JCR).

## Statistical analysis

Data were converted to txt format and imported into CiteSpaceV, GraphPad Prism 5, the Online Analysis platform of Literature Metrology (http://bibliometric.com/) and VOSviewer. Data were then analyzed cooperation network of keywords, institutions, cited reference, and authorship quantitatively and qualitatively by CiteSpaceV. GraphPad Prism version 5.0 was used to evaluate the strength and direction of the linear relations between the number of citations in the top 100 (T100) cited articles and the number of years since publication, the number of authors, the number of institutions, and the number of countries. It was also used to analyze the correlation of article citations between different databases (WoSCC and Scopus). All probability values were two-tailed, and the threshold for significance was set at $P < 0.05$. Using the Online Analysis platform of Literature Metrology, the number of included articles and the number of articles according to country, published each year were reported. The analysis also showed the number of the top 17 keywords in each year. Exhibiting the relative positions and density of nodes in a network, two dimensional knowledge maps can be produced by VOSviewer which is a computer program primarily intended to be used for mapping, analyzing and exploring different types of networks (*Su & Lee, 2010*).

# RESULTS

## Total numbers of published items

It was absolutely necessary to consider the number of published items on FMT as an index of research productivity. From 2004 to 2017, a total of 2,062 publications on FMT were identified through our search strategy in WoSCC, and 824 papers were screened out according to their titles and abstracts, 1,238 publications were excluded because they did not meet our exclusion criteria (titles uncorrelated to FMT in 809 papers, the main contents of 125 articles were irrelevant to FMT, the abstracts were inaccessible for 302 papers, and 2 papers duplicated with other articles). Moreover, 28 articles were excluded because they are non-English language. Thus, 796 articles on FMT from 2004 to 2017 were included in the analysis (Fig. 1). One article was published in 2004 ($n = 1$), and it was in 2011 ($n = 15$) and 2012 ($n = 31$) when the numbers of published articles each year began

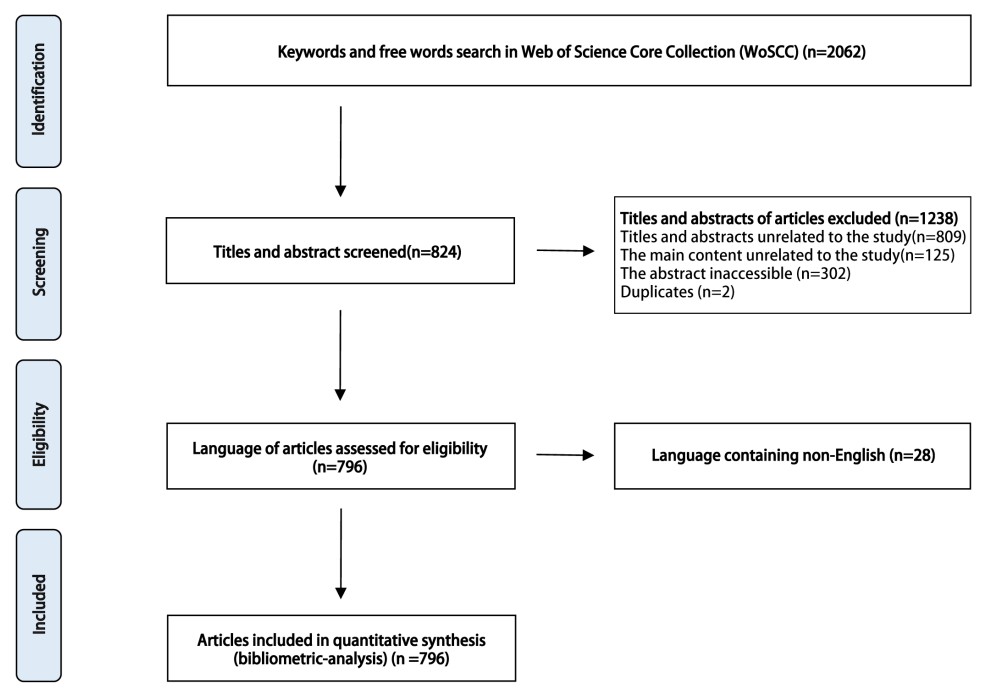

**Figure 1** Overview of article selection process.

to increase dramatically. The yearly number of published articles reached a peak in 2017 ($n = 248$) (Fig. 2A).

## Distribution by countries

The 796 publications on FMT were primarily published by ten countries/regions ($n = 722$, 90.70%) (Fig. 2B). The greatest number of publications came from the United States ($n = 363$, 50.28%), followed by China (72, 9.97%), Canada (69, 9.56%), and Germany (41, 5.68%). The United States was the only country with a dynamic growth in the number of published articles over last five years. An analysis of international cooperation was shown in Fig. 2C; the most frequent collaboration was between the United States and Canada, followed by the US and UK.

## Distribution by citations

Of the 796 selected articles, the top 100 articles ranked by the number of citations (Table 1, Table S1). The median number of citations was 95 (range 47–1,158), and three papers were cited over 500 times. The citation index (median 21.17, range 7.18–231.60) was correlated with the number of citations ($r^2 = 0.83$, $P < 0.01$) per article in the Web of Science database. In addition, the number of citations and citations index per article was strongly correlated in the Scopus database (Fig. 3: $r^2 = 0.84$, $P < 0.01$) (Fig. S1). The T100 articles were published from 2007 to 2016, with the most articles published in 2015 ($n = 22$), followed
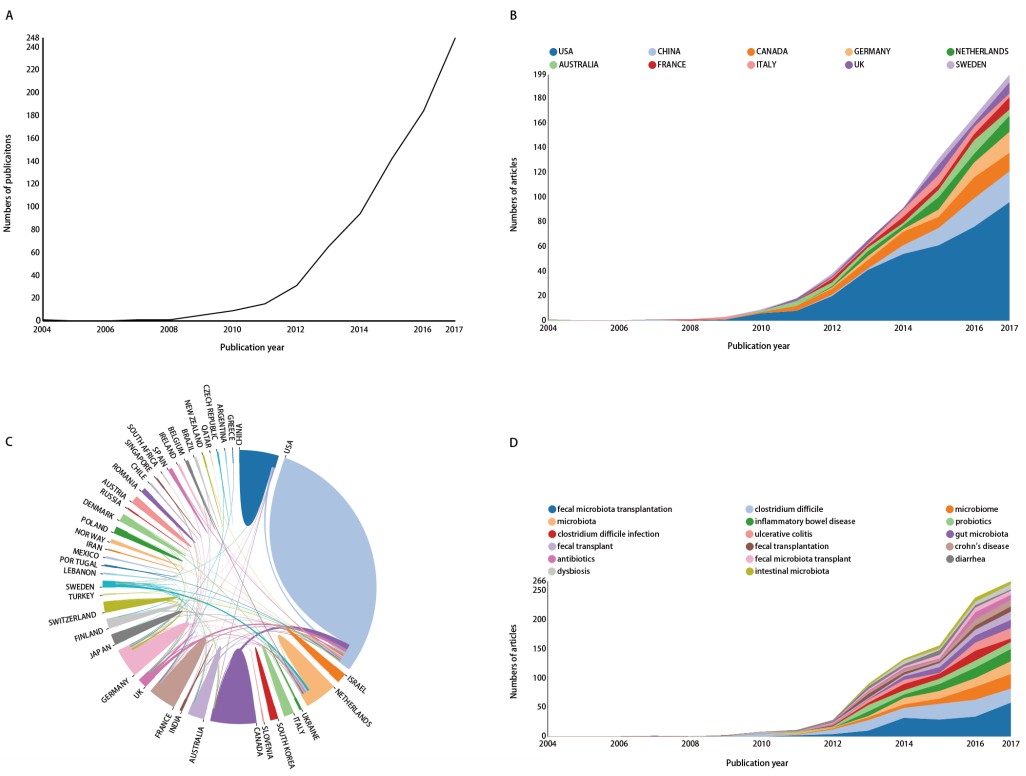

**Figure 2** The number of annual publications (A), and growth trends of countries (B), the cooperation of countries/regions (C), growth trend of keywords (D) on FMT from 2004 to 2017.

by 2013 ($n = 20$; Fig. 3). Interestingly, the number of citations in 2013 ($n = 4,136$) was the highest, followed by 2015 ($n = 2774$).

To identify factors that might influence the number of citations of the T100 articles, we analyzed possible correlations between the number of citations and the number of countries, funding, years since publication, and authors using GraphPad Prism 5 (Fig. 3). There was a strong correlation between the number of citations and the number of authors ($r^2 = 0.05$, $P < 0.05$). However, no significant correlations were present between the number of citations and the number of countries ($r^2 = 0.01$, $P = 0.46$), institutions ($r^2 = 0.01$, $P = 0.27$), and years since publication ($r^2 = 0.01$, $P = 0.27$).

## Analysis of references

Analysis of cited references is crucial to bibliometric analysis, as the scientific relevance of publications can be evaluated by the map of co-cited references. Using CiteSpaceV, the co-citation network was divided into seven co-citation clusters (Fig. 4A). The modularity $Q$ score was 0.54, and the silhouette score was 0.32. These clusters were labeled by index terms from their own citers. The largest cluster was cluster #0 labeled as 'recurrent *Clostridium difficile* infection,' followed by cluster #1, labeled as '*Clostridium difficile* infection' (Table S2).

**Table 1** Bibliometric information associated with the top 5 of the top 100 cited articles in FMT from 2004 to 2017.

| Rank | Title | Years | Times cited (WoSCC) | Citation index (WoSCC) | Time cited (Scopus) | Citation index (Scopus) |
|---|---|---|---|---|---|---|
| 1 | Van Nood E, Vrieze A, Nieuwdorp M, et al. Duodenal infusion of donor feces for recurrent *Clostridium difficile* [J]. N Engl J Med,2013,368(5):407–415. | 2013 | 1,158 | 231.60 | 1,307 | 261.40 |
| 2 | Vrieze A, Van N E, Holleman F, et al. Transfer of intestinal microbiota from lean donors increases insulin sensitivity in individuals with metabolic syndrome [J]. Gastroenterology, 2012, 143(4):913–916. | 2012 | 738 | 123.00 | 800 | 133.33 |
| 3 | Surawicz C M, Brandt L J, Binion D G, et al. Guidelines for diagnosis, treatment, and prevention of *Clostridium difficile* infections [J]. American Journal of Gastroenterology, 2013, 108(4):478–498. | 2013 | 623 | 124.60 | 641 | 128.20 |
| 4 | Gough E, Shaikh H, Manges A R. Systematic review of intestinal microbiota transplantation (fecal bacteriotherapy) for recurrent *Clostridium difficile* infection [J]. Clinical Infectious Diseases An Official Publication of the Infectious Diseases Society of America, 2011, 53(10):994. | 2011 | 473 | 67.57 | 495 | 70.71 |
| 5 | Vétizou M, Pitt J M, Daillère R, et al. Anticancer immunotherapy by CTLA-4 blockade relies on the gut microbiota [J]. Science, 2015, 350(6264):1079. | 2015 | 382 | 127.33 | 391 | 130.33 |

## Distribution by journals

The 796 articles were published by 294 journals. The top 100 were from 52 journals (Table 2, Table S3). According to the Journal Citation Reports (JCR) 2017 standards, the American Journal of Gastroenterology (2017 IF = 10.231) made contributions to the largest number of articles on FMT (10 articles, 10.00%), followed by Gastroenterology (2017 IF = 20.773; 8 articles, 8.00%), Journal of Clinical Gastroenterology (2017 IF = 2.968; 8 articles, 8.00%), and Clinical Infectious Diseases (2017 IF = 9.117; 4 articles, 4.00%).

## Distribution by authors

There were more than 3,000 authors who contributed to the publications on FMT. The network maps of the authors and co-cited authors produced by CitespaceV are shown in Figs. 4B & 4C. Kassam Z had the greatest number of articles (n = 33), followed by Khoruts A (n = 32) and Kelly CR (n = 21). The top five cited authors were Khoruts A (n = 755), by Brandt LJ (n = 720), de Vos WM (n = 557), and Nieuwdorp M (n = 509) (Tables 3 and 4).

## Distribution by institutions

The publications on FMT were from 933 institutions, and extensive cooperation network analysis was carried out between institutions (Fig. 4D). The top five institutions ranked by the number of articles published 253 articles, about 31.78% of the total. The University of Texas MD Anderson Cancer Center (n = 9, 12.44%) published the greatest number, followed by the University of Washington (n = 58, 7.23%), McMaster University (n = 34, 4.27%), Emory University (n = 32, 4.02%), and the University of Alabama at Birmingham (n = 30, 3.77%) (Table 5).

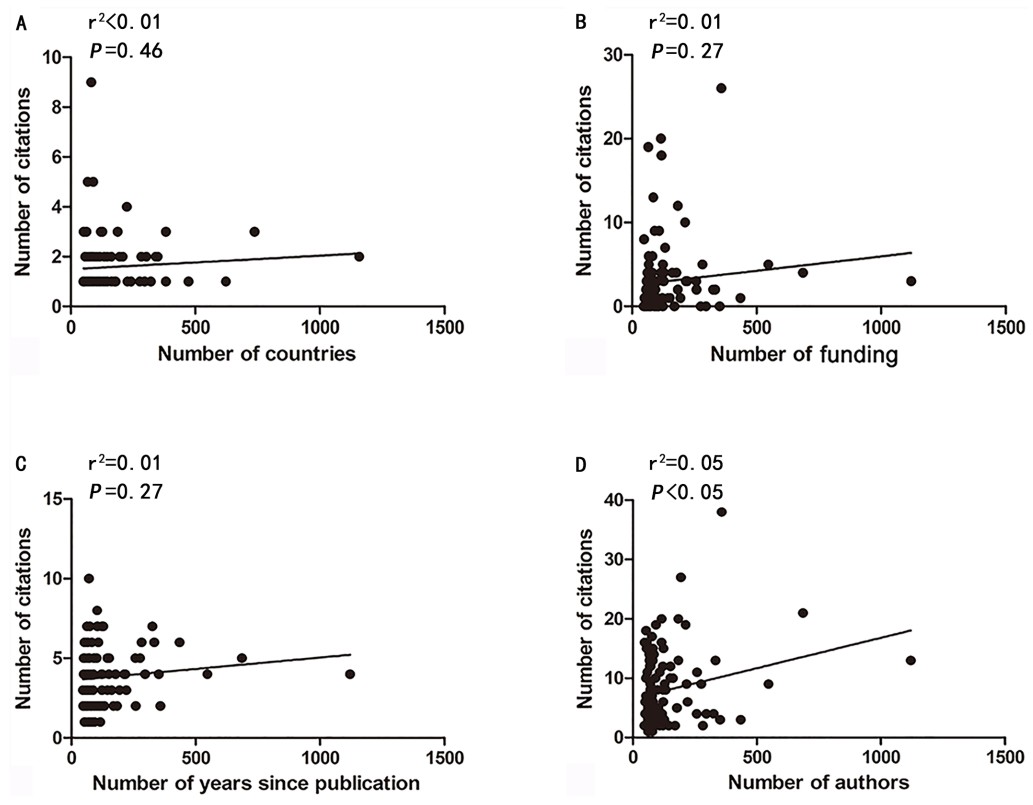

**Figure 3** Correlations between the number of citations and countries (A), institutions (B), the years since publication (C), and authors (D).

## Knowledge map of FMT research

Figure 5 illustrates a two-dimensional knowledge map with major keywords as nodes from 2004 to 2017. It shows the frequency of keywords and their relative co-occurrence with colors of different intensity to represent it as a heat map. Intense (blue) color indicates the frequency of keywords. A keyword labelled in larger font size turns up in the center of each intense color, which indicates that these keywords appeared more frequently and co-occurred with a higher number of other keywords in the literature. In addition, network visualization of keywords was shown in Fig. S2, the distance between two keywords in the visualization approximately indicates the relatedness of the keywords in terms of co-citation links. In general, the closer two keywords are located to each other, the stronger their relatedness. The strongest co-citation links between keywords are also represented by lines.

## Analysis of keywords

An approximation of research trends was found by analyzing the top 17 keywords (Fig. 2D). Almost all the keywords appeared rising and falling fluctuations but an ascendant trend. The keywords of 'fecal microbiota transplantation' and '*Clostridium difficile*' were the first and second most frequent in the last five years of the study period. Except for searching
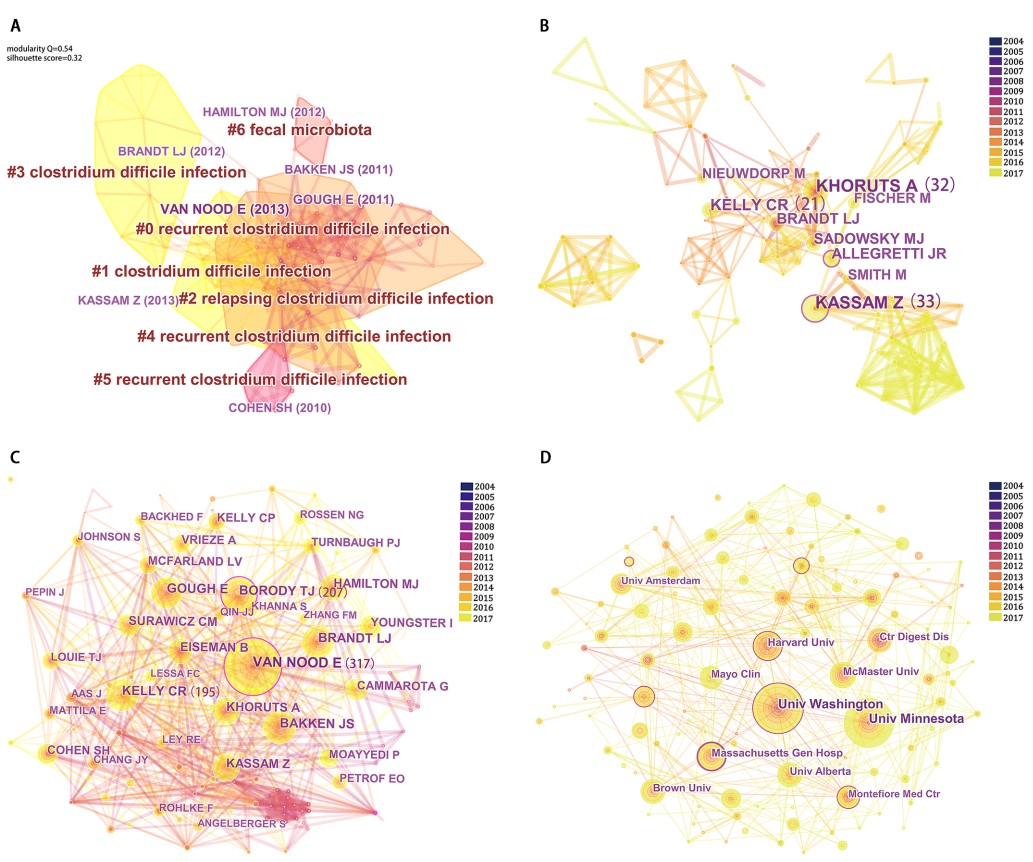

**Figure 4** Reference co-citation map (A), Network map of authors (B), Network map of co-cited authors (C) and Network map of institutions (D) contributed to publications on FMT from 2004 to 2017.

**Table 2** The top 5 journals of the top 100 articles ranked by the number of citation contributed to publications on FMT from 2004 to 2017.

| Rank | Journal | Country | No. of articles | IF 2017 |
|---|---|---|---|---|
| 1 | AMERICAN JOURNAL OF GASTROENTEROLOGY | the US | 10 | 10.231 |
| 2 | GASTROENTEROLOGY | the US | 8 | 20.773 |
| 3 | JOURNAL OF CLINICAL GASTROENTEROLOGY | the US | 8 | 2.975 |
| 4 | CLINICAL INFECTIOUS DISEASES | the US | 4 | 9.117 |
| 5 | CELL | the US | 3 | 31.398 |

keywords which were 'fecal microbiota transplantation, 'fecal transplantation', and 'facal transplant', the three most frequent keywords were 'Clostridium difficile', 'microbiome', and 'inflammatory bowel disease'. These three words are also the main therapeutic directions of all FMT researches in the world.

The top seven keywords with the strongest citation bursts were extracted by CiteSpaceV (Table 6). The blue line represents the time interval and the red line represents the duration of a burst keyword, suggesting the beginning and the end of the time interval of each burst.

**Table 3   The top five authors ranked by the number of articles.**

| Rank | Authors | No. of articles | Total citations | First | Citations of first | Correspond | Citations of correspond |
|---|---|---|---|---|---|---|---|
| 1 | Kassam, Z | 38 | 325 | 4 | 186 | 1 | 7 |
| 2 | Khoruts, A | 33 | 755 | 6 | 153 | 9 | 202 |
| 3 | Kelly, CR | 24 | 375 | 5 | 243 | 8 | 297 |
| 4 | Sadowsky, MJ | 23 | 405 | 2 | 0 | 6 | 148 |
| 5 | Allegretti, JR | 22 | 22 | 5 | 4 | 1 | 0 |

**Table 4   The top five authors ranked by the number of citations.**

| Rank | Authors | No. of articles | Total citations | First | Citations of first | Correspond | Citations of correspond |
|---|---|---|---|---|---|---|---|
| 1 | Khoruts, A | 33 | 755 | 6 | 153 | 9 | 202 |
| 2 | Brandt, LJ | 20 | 720 | 8 | 271 | 9 | 342 |
| 3 | de Vos, WM | 8 | 557 | 0 | 0 | 0 | 0 |
| 4 | Nieuwdorp, M | 18 | 509 | 0 | 0 | 9 | 171 |
| 5 | Zoetendal, EG | 4 | 472 | 0 | 0 | 0 | 0 |

**Table 5   The top five institutions ranked by the number of articles contributed to publications on FMT from 2004 to 2017.**

| Rank | Institution | No. of articles | No. of citations |
|---|---|---|---|
| 1 | University of Minnesota, Gemini, Minnesota, the US | 99 | 1999 |
| 2 | University of washington, Seattle, Washington, the US | 58 | 1136 |
| 3 | McMaster University,ilton, Ontario, Canada | 34 | 1099 |
| 4 | Emory University, Atlanta, GA, the US | 32 | 119 |
| 5 | University of Alabama Birmingham, Birmingham, Alabama, the US | 30 | 58 |

The top keyword was 'flora' (10.56, 2011–2014), followed by 'bacteriotherapy' (7.69, 2009–2013) and 'diarrhea' (7.02, 2011–2013).

There were 722 different keywords in the 796 publications. All keywords with the same meaning were merged into one keyword. After data standardization, 481 keywords were selected as core keywords. Among them, 59 keywords appeared more than three times whose frequency of occurrence were 70.24% so that they were selected as core keywords. The classification results in Table 7 are shown in four domains: disease, nosogenesis, trial, and therapy. Among the 59 core keywords, the highest percentage was in the disease domain (40.49%), followed by the therapy domain (41.44%), the nosogenesis domain (14.63%) and the trial domain (3.77%). The transplantation topic in the therapy domain contained the highest percentage (33.30%) of core keywords, which were 'fecal microbiota transplantation' 'microbiome' 'stool transplantation' 'bacterial consortium transplantation' 'bacteriotherapy' 'gut microbiome transplantation' and 'transplantation'. In the disease domain, infection topics (16.79%) occurred most frequently. Microbiota topics (11.13%)
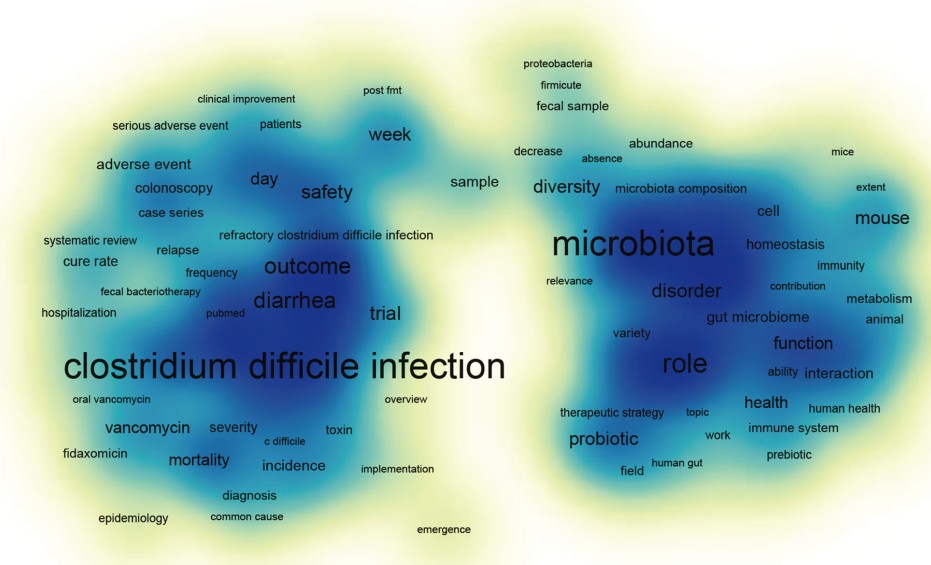

**Figure 5  Visualization of two-dimensional knowledge map of keywords for the complete dataset.**

occurred most frequently in the nosogenesis domain. The subject topic in the trial domain accounted for 2.42%.

## DISCUSSION

### Total number of published items

This was the first bibliometric analysis of FMT. The yearly number of publications rapidly increased from one article in 2004 to 248 articles in 2017, with an average annual increase of 17.35% over the last 5 years of the study period. Compared with the total scientific output in the WoSCC database, its average annual growth rate was 3.84% between 2004 and 2017. The increasing incidences of CDI and IBD might explain the increase in the numbers of publication from 2004 to 2017 (*Rohlke & Stollman, 2012*; *Kelly, De & Jasutkar, 2012*). In addition, the increasing number of output index in the WoSCC database might have also made contribution to the increase in the number of publications. A conclusion can be drawn from the rapid development of FMT, whose research was only in its infancy but promising and potential.

### Country of origin and institutions

The top 10 countries/regions in which FMT studies were performed accounted for for 90.70% of the total number of publications. China was the only developing country among these 10 countries/regions, indicating that China has made great progress in the FMT research recently. It is not surprising that developed countries produced the greatest numbers of publications (*Van Rossum et al., 2010*; *Halpenny et al., 2010*). The United States, which produced 363 publications (45.60%), demonstrated a leading role in FMT research.

**Table 6 Core Keywords related to FMT from 2004 to 2017.**

| Domain | Topic | Percentage within core keywords, % | Frequency of keyword occurrence (n) |
|---|---|---|---|
| **Disease domain (total, 40.49%)** | Infection | 16.79 | Clostridium difficile infection (171), infection (5), bacterial infections (4), refractory clostridium difficile infection (4), fulminant clostridium difficile infection (3) |
| | Inflammatory | 13.73 | Inflammatory bowel disease (66), ulcerative colitis (38), crohn's disease (23), colitis (11), inflammation (8), pseudomembranous colitis (7) |
| | Diarrhea | 5.48 | Antibiotic-associated diarrhea (26), diarrhea (16), recurrent clostridium difficile infection (16), infectious diarrhea (3) |
| | Metabolism | 2.6 | Obesity (12), metabolic syndrome (9), diabetes (4), metabolism (4) |
| | Functional disease | 1.26 | Irritable bowel syndrome (11), functional bowel disease (3) |
| | Emotion | 0.63 | Depression (4), anxiety (3) |
| **Nosogenesis domain (total, 14.63%)** | Microbiota | 11.13 | Gut microbiota (49), microbiota (40), intestinal microbiata (27), bifidobacterium (5), lactobacillus (3) |
| | Environment | 2.69 | Dysbiosis (17), gut (4), rain-gut axis (3), gastrointestinal (3), gut permeability (3) |
| | Immunity | 0.81 | Immunomodulation (9) |
| **Trial domain (total, 3.77%)** | Subject | 2.42 | Children (13), germ-free (5), germ-free animals (3), mathematical modeling (3), mouse models (3) |
| | Method | 1.35 | 16s rRNA analysis (8), metagenomics (4), clinical trial (3) |
| **Therapy domain (total, 41.44%)** | Transplantation | 33.3 | Fecal microbiota transplantation (301), microbiome (41), stool transplantation (9), bacterial consortium transplantation (7), bacteriotherapy (6), gut microbiome transplantation (4), transplantation (3) |
| | Food | 4.13 | Probiotics (33), prebiotic (8), diet (5) |
| | Antibiotics | 2.87 | Antibiotic therapy (13), fidaxomicin (6), vancomycin (6), metronidazole (4), infection control (3) |
| | Fibroptic endoscopy | 0.81 | Colonoscopy (9) |

**Table 7 The top 7 Keywords with the Strongest Citation Bursts on FMT from 2004 to 2017.**

| Keywords | Year | Strength | Begin | End | 2004–2017 |
|---|---|---|---|---|---|
| flora | 2004 | 10.5588 | **2011** | 2014 | ▭▭▭▭▭▭▭▭▭▭▭▭▭ |
| bacteriotherapy | 2004 | 7.6856 | **2009** | 2013 | ▭▭▭▭▭▭▭▭▭▭▭▭▭ |
| diarrhea | 2004 | 7.0221 | **2011** | 2013 | ▭▭▭▭▭▭▭▭▭▭▭▭▭ |
| case series | 2004 | 6.2972 | **2011** | 2013 | ▭▭▭▭▭▭▭▭▭▭▭▭▭ |
| antibiotic associated diarrhea | 2004 | 5.4075 | **2010** | 2013 | ▭▭▭▭▭▭▭▭▭▭▭▭▭ |
| coliti | 2004 | 5.0701 | **2010** | 2013 | ▭▭▭▭▭▭▭▭▭▭▭▭▭ |
| enterocolitis | 2004 | 4.3316 | **2010** | 2013 | ▭▭▭▭▭▭▭▭▭▭▭▭▭ |

This might be related to the high incidence of CDI and IBD in the United States (*Orenstein, Griesbach & Dibaise, 2013*). It might also be related to the financial resources devoted to scientific study in the United States (*Zhao et al., 2016*). In addition, the United States played a key role in promoting international cooperation, with the strongest international

cooperation between the United States and Canada, followed by between the United States and the United Kingdom (Fig. 2C). As regard to the cooperation of institution, the top five institutions were ranked by the number of articles, among which four institutions subordinated to the United States. This conclusion was consistent with the dominance of the United States. A conclusion could be drawn that the increasing incidence of CDI and IBD has promoted research of FMT in this field.

## Citation count and possible factors influencing citations

Based on the clarity of network structure and clustering, two indexes were proposed by CitespaceV: the modularity $Q$ score and the mean silhouette score. Modularity and silhouette metrics provided useful quality indicators of clustering and network decomposition. A low modularity means that a network can't be reduced to a cluster with clear boundaries, while a high modularity suggests a well-structured network. The silhouette metric is helpful for estimate the uncertainty involved in identifying the nature of a cluster (Chen, Ibekwe-Sanjuan & Hou, 2010). The co-citation network was divided into 7 co-citation clusters with a modularity of 0.54 and a silhouette of 0.32. This result suggested, the inter-cluster connections are considerable, but not overwhelming and are diverse and heterogeneous.

Citation counting, a proxy measure of research quality, can help authors understand the characteristics inherent in highly cited studies and provide a new perspective on specific areas (Yan et al., 2011; Fu & Aliferis, 2008). However, there has been no citation analysis of FMT until now. Of 796 selected articles, we focus on the top 100 articles arranged by the number of citations. Interestingly, there was no output in 2017. It's likely that 'older' articles have a longer citable period and attained more citations, and as such accumulative citation frequency will be higher (Liu et al., 2016). To further examine this, we assessed the correlation between years since publication and citation count. However, no obvious correlation was found. In addition, we replaced citation count with citation index to decrease the effect of publication time. Citation counts are quantitative, they can be added, subtracted, normalized, and ploted, satisfactorily, citation index based on these counts abound, the ubiquitous Hirsch or h-index being the most prominent (Will, 2014), and the h-index is an author-level metric that attempts to measure both the productivity and citation impact of the publications of a scientist or scholar (Prathap, 2012). The results indicated that citation count and citation index exhibited a substantial correlation, indicating that publication time has very little influence on citations. In addition, we analyzed the citations of the top 100 publications in Scopus. A strong relationship was found between citation index in WoSCC and citation index in Scopus, so does the relationship between citations and citation index in Scopus, which ruled out the impact of the database on citations.

## Journals

The IFs of journals were one of the strongest indicators for citations in some way owing to the attraction of high IF journals to the scientific community (Garfield, 2003). The top-cited articles are usually published in high IF journals. The top 100 articles ranked by the number of citation were published in 52 journals, with the exception of one, because

its IF was not found in the JCR. The median IF of these publications was 6.96, and the Ifs of 18 (35.29%) publications were greater than 10.00. These results suggested that it was challenging to publish articles on FMT in high IF journals. Meanwhile, it implied the quality of output in this subject area.

## Authorship

Ranked by the number of articles they owned, the top five authors identified in this analysis published at least 22 articles. They were consequently regarded as 'prolific authors'. The top five authors contributed to at least 472 citations. Surprisingly, there was one author in both analyses where this author not only did well in the number of publications but also the quality of publications. The co-cited authors who had at least 200 co-citation counts, included Van Nood E and Borody TJ. Although neither of them belonged to the category of prolific authors, they played pivotal roles in FMT research; particularly, Van Nood E, who was also the first author of the article with the highest number of citations.

## Knowledge map of FMT research

A total of 243 items were divided into four clusters, cluster 1, 2, 3, and 4 comprised of 122, 144, nine and one items, respectively (Fig. 5 & Fig. S2). In the item density visualization, each point has a color that indicates the density of items at that point. Colors in the map range from blue to yellow, the larger the number of items (i.e., *clostridium difficile* infection) in the neighborhood of a point and the higher the weights of the neighboring items, the closer the color of the point is to blue. In the network visualization, items are represented by their label and by default also by a circle. The size of the label and the circle of an item is determined by the weight of the item. The higher the weight of an item, the larger the label and the circle of the item. The color of an item is determined by the cluster to which the item belongs, for example, microbiota in green belong to cluster 2. Lines between items represent links, meanwhile, the distance between two keywords in the visualization approximately indicates the relatedness of the keywords in terms of co-citation links. In general, the closer two keywords are located to each other, the stronger their relatedness. The strongest co-citation links between journals are also represented by lines.

## Keywords and research fields

In recent years, there has been great interest in the use of FMT for therapy of gastrointestinal and non-gastrointestinal diseases, mainly for CDI and IBD. In most countries/regions, the incidence of IBD is increasing or at a relatively high stable level. Furthermore, IBD patients have a nearly threefold higher risk for CDI compared with the general population (*Rodemann et al., 2007*). Nevertheless, some patients became refractory to standard therapy and suffered from a poor quality of life. Evidence suggested that FMT was a potential, effective and safe therapy for recurrent, refractory, or severe CDI and IBD, at least when standard treatments had failed (*Anderson, Edney & Whelan, 2012*; *Kelly et al., 2014*). The goal of this bibliometric analysis was to provide an overview of FMT's research, and guide future studies.

According to the 59 core keywords identified, 4 domains were classified. The disease domain had the highest percentage and the trial domain the lowest. Within the disease

domain, the topics of infection and inflammatory have the highest ratio. In this regard, it might relate to the therapy principle of FMT. Evidence has suggested that there is a reduced diversity of luminal microbiota and increased mucosal adherent bacteria in IBD and CDI (*Nagalingam & Lynch, 2012*). Changes in gut microbiota led to a serious imbalance of host physiology and immune homeostasis, which ultimately results in infection and inflammation. *Brandt et al. (2012)* noted that FMT was a promising therapeutic strategy because it manipulated the microbiota, based on gastrointestinal microbiota's role in driving disorders. Interestingly, the topic of emotion taken up a certain proportion, which was an indication that FMT could be used to treat mental illness.

In the therapy domain, the topic of transplantation occupied the largest portion and other major treatments of IBD and CDI were also listed. In general, the standard therapy for CDI and IBD included steroids, aminosalicylates, immunosuppressants, and various biological therapies, most of which are not effective and place a heavy economic burden on patients (*Talley et al., 2011*; *Kappelman et al., 2008*). Some patients become refractory to standard management and suffer significant adverse side effects with a poor quality of life (*Mcfarland, 2005*). *Konijeti et al. (2014)* indicated that FMT was the most cost-effective initial strategy for management of recurrent CDI. FMT can thus reduce the financial burden of patients, and result in a substantial increase in quality of life (*Merlo, Graves & Connelly, 2016*).

The domains mentioned above clarified research hotspots clear, and burst keywords can be considered indicators of research frontiers over time (*Yin, Chen & Li, 2016*). The blue line represents the time interval and the red line represents the duration of a burst keyword. There were seven keywords with the strongest citation bursts. After taking similar burst keywords into consideration, they were divided into three categories: (i) Microbiota: the presence of normal, healthy, intestinal microbiota was now considered to offer protection against CDI and IBD. However, the repeated use of immunosuppressants and other therapy with drugs severely disrupted the normal gut microbiota, which always led to the recurrence of CDI and IBD. Instead, FMT allowed the rapid reconstitution of a normal composition of microbial communities. The wonderful effect and desperate need in the patients would drive the development of FMT in the next few years (*Hamilton et al., 2012*). (ii) Diarrhea: antibiotic-associated colitis caused by *Clostridium difficile* was the most common cause of hospitalization for diarrhea (*Cohen et al., 2015*). It was caused by disrupting the normal gut flora and led to dysbiosis that enables *Clostridium difficile* colonization of the patients' gut. Satisfactorily, FMT, a promising therapy for *Clostridium difficile*-associated diarrhea, has the high cure rate of diarrhea (*Kelly et al., 2014*; *Gough, Shaikh & Manges, 2011*). However, some problems should be further solved and perfected, such as whether FMT should be used as a first-line therapy for the patients with *Clostridium difficile*-associated diarrhea. (iii) Case series: there are many different treatments for CDI and IBD, but none of the treatments are proved to be very useful. In order to find the solution that worked best, it was necessary to analyze representative cases. *Rubin et al.( 2013)* found the effectiveness of FMT in the therapy of CDI and IBD from a case series of 75 FMT course. This gave the future researchers a clue that case series would be an appropriate way.

There are some limitations to the current study. Non-English publications were excluded and as landmark articles published in other languages were not considered. The study only focused on the publications in WoSCC database, and the exclusion of other databases, such as PubMed and Scopus, may have produced slightly different results. Nevertheless, WoSCC is a comprehensive and popular Web database in the field of scientometrics (*Miao et al., 2017*).

## CONCLUSIONS

In conclusion, based on the detailed analysis of the distribution and changes in countries, citations, references, journals, authorship, institutions and keywords, the results of this research and analysis indicate that FMT is an area of very active research. This analysis using bibliometric methods provides a solid overview of current FMT research and may help in providing guidance for further studies.

**Abbreviations**

| | |
|---|---|
| **FMT** | fecal microbiota transplantation |
| **CDI** | *Clostridium difficile* infection |
| **IBD** | inflammatory bowel disease |
| **SCI-E** | the Science Citation Index Expanded |
| **WoSCC** | the Web of Science Core Collection |
| **T100** | the top 100 |
| **JCR** | the Journal Citation Reports |

## ACKNOWLEDGEMENTS

The authors would like to thank editors and the anonymous reviewers for their valuable comments and suggestions to improve the quality of the paper.

### Funding

This study was supported by the Undergraduate Innovation and Entrepreneurship Training Program (No.201712121157). The funders had no role in study design, data collection and analysis, decision to publish, or preparation of the manuscript.

### Grant Disclosures

The following grant information was disclosed by the authors:
The Undergraduate Innovation and Entrepreneurship Training Program: No.201712121157.

### Competing Interests

The authors declare there are no competing interests.

## Author Contributions

- Yan Li conceived and designed the experiments, performed the experiments, authored or reviewed drafts of the paper, approved the final draft.
- Ziyuan Zou conceived and designed the experiments, contributed reagents/materials/analysis tools.
- Xiaohui Bian contributed reagents/materials/analysis tools, prepared figures and/or tables, authored or reviewed drafts of the paper.
- Yushan Huang performed the experiments, analyzed the data, contributed reagents/materials/analysis tools.
- Yanru Wang authored or reviewed drafts of the paper.
- Chen Yang contributed reagents/materials/analysis tools, authored or reviewed drafts of the paper.
- Jian Zhao analyzed the data, contributed reagents/materials/analysis tools.
- Lang Xie conceived and designed the experiments, authored or reviewed drafts of the paper.

## Data Availability

Raw data are available in the Supplemental Files.

## Supplemental Information

Supplemental information for this article can be found online at http://dx.doi.org/10.7717/peerj.6411#supplemental-information.

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
