# Peer review of "Fecal microbiota transplantation research output from 2004 to 2017: a bibliometric analysis"

_PeerJ, doi:10.7717/peerj.6411_

## Round 0.1 · original submission · Major Revisions

Thank you very much for your review. Although the content experts feel you have covered the topic well, reviewers with considerable experience in conducting reviews feel the there is not enough methodological detail concerning how the review was conducted to warrant publication at this time. Additionally, at least one reviewer indicated your submission could greatly benefit from editing by a native language speaker, although admittedly you have done much better writing a paper in English than this editor would do writing a paper in Mandarin.

Reviewer 1 ·

Basic reporting

It contained a relevant interpretation of data. It confirmed to the style and format of PeerJ.

Experimental design

Methods were enough to draw results to reply to the purpose of the study.
Network analysis was also good.

Validity of the findings

Results and discussion was well-described. the conclusion is simple and lucid.

Additional comments

I recommend the revision as follows:

line 70 Core Collection (WoSCC) of Thomson Reuters
-->Core Collection (WoSCC) of Clarivate Analytics

106 number of national articles published each year were reported
-->number of articles according to country, published each year were report.

Otherwise, description was good.

Reviewer 2 ·

Basic reporting

1. The English expression should be clarified more to be clearly understood by the international audience.
- Line 101 ~103, Line 328-330

2. Line 70/Line 394 : Thomson Reuters - Company name was changed as Clarivate Analytics(https://clarivate.com/)

3.Line 71: WOS is not a public database. It needs a subscription. It needs to be revised.

Experimental design

1. Lin 85 : It is difficult to understand why you exclude the articles which cannot be acquired from WoS. As you mentioned in Line 83, most abstracts could be acquired from PubMed or the publihsers' homepage.

2. Line 112-116: It would be good to explain more clearly based on the overview of article selection process(Figure 1)

3. Line 113 : On the text, you mentioned 1,277 papers were excluded. However, on the Figure 1, you wrote 1,238. Please correct it.

4. Line 103-104 : It would be good to explain about Scopus search keywords and search date.(law data also could be uploaded)

Validity of the findings

1.Line 225-228 : Needs to be added references to validate this sentence

2. Line 246-247 : Needs to explain why you select only top 100 highly cited papers

3. Line 251 : For not familiar with bibliometric analysis users, it would be good to explain the difference of "citation count" and "citation index".

4. Line 261-265 : Needs to write more clearly and explain more detail why you get this result(Line 265)

Additional comments

This paper examines an interesting topic using bibliometric analysis.
Although this paper shows some interesting findings, description of methodology is a little bit unclear to convince study findings related to the stated study purposes. For instance, you should mention more Figure 4 because it's quite a good visual diagram. In addition, you could explain more the differences between each bibliometric tools. It is good to add some benefits why you used bibliometric analysis, not a systematic review or meta-analysis.

Reviewer 3 ·

Basic reporting

Figure 3 does not make a good sense, especially the linear line. Other reporting looks okay

Experimental design

Data extraction is okay but need further rigorous and in-depth analyses as outlined in the previous section

Validity of the findings

okay; most of the parts of the manuscript cover the basic reporting - it requires some further sophisticated data analysis to reveal any unknown findings

Additional comments

It seems the authors made a significant effort in this manuscript. However, most of the analyses they conducted are at very basic level. I would recommend the authors to follow these two articles as guidelines for further research analyses

Longitudinal trends in global obesity research and collaboration: a review using bibliometric metadata
A Khan, N Choudhury, S Uddin, L Hossain, LA Baur
Obesity Reviews 17 (4), 377-385


A Framework to Explore the Knowledge Structure of Multidisciplinary Research Fields
S Uddin, A Khan, LA Baur
PLOS ONE 10 (4), e0123537

·

Basic reporting

• English language grammar and/or punctuation is frequently incorrect or unclear. See lines 44, 64, 94, 99, 114, 115, 118, 128 for relevant examples.
• Appropriate literature references to establish background research are sometimes omitted. See lines 57 through 60 for a good example.
• Words/phrases unnecessarily repeated lead to awkward sentences. See lines 65, 47-49 for relevant examples.
• Verb tenses and references to interactions are often incorrect. See lines 158 [American Journal of Gastroenterology… made contributions], 171 [extensive cooperation was carried out between institutions,], 180 [almost all the keywords appeared rising and falling fluctuations..] 212 [the increasing number of output indexed], 255 [was found, so does the relationship between].

Background: Unclear introduction as stated by line 26 [there has been no bibliometric analysis of FMT]. Analysis of what aspect of this literature and how does this relate to prominence of the literature? The meaning of “prominence” is vague and undefined. The reason for conducting a bibliometric analysis of an emerging therapy (FMT) is vaguely implied but not stated explicitly.
Methods: Line 28 states “cooperation of publications”. This is a confusing and grammatically incorrect sentence.
Results: Lines 33-36. Unclear statements of relationship and significance of keywords and domains.
Conclusions: Line 38 “Relationship between research performance of FMT” to “investigators master the trend” illogical as stated and grammatically incorrect use of English language active tense and transition. Closing sentence states a foregone conclusion “..which is also an ongoing hotspot of research”.

Experimental design

no comment

Validity of the findings

While there appears to be solid research presented the lack of idiomatic English and of clear explanations when needed make the article difficult to read. Native English readers could struggle through but be frustrated by incorrect grammar, sentence structure, and words that poorly describe meaning of the statement.

The conclusion is poorly stated and relates the purpose of the study to a foregone conclusion (e.g. that such studies aid undefined investigators)

Thorough editing of these elements would be required before considering publication.

Additional comments

The authors of this article have been able to generate a substantial amount of data appropriately using the tools selected for a bibliometric analysis. However, the errors and omissions as described in the previous review categories lead me to conclude that the paper is not acceptable for publication in the current form.

---

## Round 0.2 · Minor Revisions

The reviews received by the original reviewers were somewhat disparate so this editor chose to find an additional reviewer. It is somewhat concerning that one of the original reviewers (#3) recommended rejected your manuscript. In revising your manuscript you are implored to pay attention to Reviewer 3's previous comments who recommended rejection because, among other things, "The author(s) do not follow my suggestions on the previous draft." Please pay attention to *all* the reviewer's comments and suggestions and implement them when possible. Thank you very much.

Reviewer 1 ·

Basic reporting

no comment

Experimental design

no comment

Validity of the findings

no comment

Additional comments

Manuscript was revised according to reviewers' comments appropriately.

Reviewer 3 ·

Basic reporting

Made some effort, but not enough

Experimental design

Almost same as like the initial draft

Validity of the findings

Not quite sure

Additional comments

The author(s) do not follow my suggestions on the previous draft.

·

Basic reporting

I am extremely pleased with the edits made to conform to conventional English language grammar and punctuation. Clearly the authors diligently worked to revise the problematic language and to clarify the meaning that was often unclear in the original submission.
Literature references that were added soundly supported the statements made to provide context to the meaning of the research. Due to the meaningful revisions, the questions that I posed regarding the conclusions drawn from previous studies are satisfactorily answered.
Figures are exceptionally striking and the enhancements highlight the most interesting results of the mapping.Tables are well organized and presented in such a way that subsequent studies will easily update the publication patterns. The highlighting used in Table 7 is particularly effective and I look forward to seeing followup data that will capture relevant terminology as the research base advances.

One slight edit needed due to improper tense in line 289: "These results suggested that it was challenging to published articles on FMT"

Experimental design

no comment

Validity of the findings

no comment

Additional comments

Revision very well done. I commend the authors for the attention to recommended editing. I believe that this research is now presented in such a way that it is likely to significantly build the understanding of the status of research into FMT on an international scale.

·

Basic reporting

• Lines 58-60: Run-on sentence. Also, what evidence do you have that systematic reviews fail to include all relevant research? What evidence do you have to prove publication bias?
• Line 60-61: Provide reference.
• Lines 70, 77: Clarify why you chose 2004 as your starting point.
• Line 85: Correct grammar.
• Line 97-98: Correct grammar. Maybe say, “…impact factors were also recorded using the 2017 edition of Journal Citation Reports (JCR)”.
• Line 101, 102: Delete extra comma. Remove underline from “and VOSviewer”.
• Line 103: the word “analyze” is used twice; replace one iteration with synonym.
• Line 139-140: Add space between “top” and “100”. Also, what exactly are you trying to say here? Please clarify your statement.
• Line 287: Correct grammar, say instead, “with the exception of one”
• Line 371: Correct grammar. Say instead, “…. the inclusion of other databases, such as PubMed and Scopus, may have produced…”

Experimental design

• Line 78: A librarian should have performed the search to ensure that you didn’t miss any relevant articles that may have used different synonyms. You mentioned other possible keywords resulting from your analysis in lines 221-223.
• Line 86-87: Why was full-text downloaded from databases other than WoS? We don’t need to know where/how you got full-text.
• Line 120: Explain things better and correct grammar: “….a total of 2,062 publications were identified through our search strategy….. 1,238 publications were excluded because they did not meet our exclusion criteria (misleading titles/abstracts; FMT not main topic; abstract inaccessible; duplicates)…. 28 additional articles were excluded because they were non-English language.)

Validity of the findings

• Line 134: What do you consider “dynamic growth”?
• Line 266 (and earlier): Did you include or exclude self-citations? You should mention this.
• Line 276: H-index is the impact of an individual author, not a publication, so it is confusing/misleading to mention here without more context.
• Line 284-286: You say that the top-cited articles are usually published in high IF journals….but this is a catch-22…. Maybe the articles are not higher quality but just get cited more because the journal is more ‘prestigious’ but this is a self-sustaining cycle.
• Line 303: Define what you mean by “items”. Keywords? What about FMT?
• Line 368: Additional study limitations include non-comprehensive search strategy (could have missed relevant articles that used other synonyms for each concept); articles without abstracts were excluded.

---

## Round 0.3 · accepted · Accept

Thank you very much for working diligently towards a much-improved product. I am happy to approve the submitted manuscript now. Happy New Year.